# The CNS and the Brain Tumor Microenvironment: Implications for Glioblastoma Immunotherapy

**DOI:** 10.3390/ijms21197358

**Published:** 2020-10-05

**Authors:** Fiona A. Desland, Adília Hormigo

**Affiliations:** Icahn School of Medicine at Mount Sinai, The Tisch Cancer Institute, Department of Neurology, Box 1137, 1 Gustave L. Levy Pl, New York, NY 10029-6574, USA; fiona.desland@icahn.mssm.edu

**Keywords:** CNS immunity, tumor microenvironment, glioblastoma, immunotherapy

## Abstract

Glioblastoma (GBM) is the most common and aggressive malignant primary brain tumor in adults. Its aggressive nature is attributed partly to its deeply invasive margins, its molecular and cellular heterogeneity, and uniquely tolerant site of origin—the brain. The immunosuppressive central nervous system (CNS) and GBM microenvironments are significant obstacles to generating an effective and long-lasting anti-tumoral response, as evidenced by this tumor’s reduced rate of treatment response and high probability of recurrence. Immunotherapy has revolutionized patients’ outcomes across many cancers and may open new avenues for patients with GBM. There is now a range of immunotherapeutic strategies being tested in patients with GBM that target both the innate and adaptive immune compartment. These strategies include antibodies that re-educate tumor macrophages, vaccines that introduce tumor-specific dendritic cells, checkpoint molecule inhibition, engineered T cells, and proteins that help T cells engage directly with tumor cells. Despite this, there is still much ground to be gained in improving the response rates of the various immunotherapies currently being trialed. Through historical and contemporary studies, we examine the fundamentals of CNS immunity that shape how to approach immune modulation in GBM, including the now revamped concept of CNS privilege. We also discuss the preclinical models used to study GBM progression and immunity. Lastly, we discuss the immunotherapeutic strategies currently being studied to help overcome the hurdles of the blood–brain barrier and the immunosuppressive tumor microenvironment.

## 1. Introduction

Glioblastoma (GBM), a World Health Organization (WHO) grade IV glioma, is the most common and malignant primary brain tumor in adults. Despite aggressive standard and targeted therapy, patients have a 14-month median survival following diagnosis. The survival timeline is slightly extended for the patients whose tumors express methylation of the promoter of the DNA repair gene, O-6-methylguanine-DNA methyltransferase (MGMT) [1]. The epigenetic silencing of this gene renders the tumor more susceptible to alkylating chemotherapies [1]. Not only is GBM an exceedingly infiltrative tumor but it also has extensive intra-tumoral heterogeneity at both the cellular and molecular levels [2,3,4], driving its high rate of therapeutic resistance and the inevitability of recurrence.

While most of these tumors arise de novo, about 10% arise secondarily from a low-grade glioma and are typically more likely to occur in younger patients [5,6]. Gliomas have been clinically graded based on their histopathological characteristics [7]; however, work in recent years has incorporated tumor gene expression, gene mutation, and epigenetic signatures into a patient’s tumor classification, prognosis, and treatment response [8,9,10,11]. Work from the Cancer Genome Atlas (TCGA) and others has molecularly categorized GBM tumors into the proneural, classical, and mesenchymal subtypes—marked predominantly by mutations causing platelet-derived growth factor receptor alpha (PDGFRa) activation, epidermal growth factor receptor (EGFR) activation, and neurofibromin 1 (NF1) deletions, respectively [11]. These subtypes, however, can co-exist to varying degrees within one lesion. In addition, mutations in isocitrate dehydrogenase 1 and 2 (IDH1/2) are now part of standard tumor genotyping. IDH1 and IDH2 help to initiate tumorigenesis by converting α-ketoglutarate (α-KG) to the oncometabolite 2-hydroxyglutarate (2-HG) [12]. Mutations in these genes are a discerning marker for secondary GBM and, most importantly, are associated with a better prognosis for patients [6,13].

Beyond prognostication, molecular classification also helps refine systemic targeted therapies that can be used in conjunction with standard care treatment. The typical GBM treatment regimen includes tumor resection, radiotherapy with concurrent chemotherapy with temozolomide, followed by adjuvant temozolomide chemotherapy [14]. While several potential targeted therapies have emerged over the years [15], only bevacizumab, an anti-angiogenic agent, has been approved by the U.S. Food and Drug Administration (FDA). As such, little headway has been made in the last 30 years in significantly reducing GBM morbidity and mortality. Fortunately, the explosion in immunotherapy has opened new treatment avenues for patients with aggressive gliomas.

Infiltration of blood-derived immune cells into the central nervous system (CNS) is typically limited to occasions when the blood–brain barrier (BBB) is disrupted, as is readily seen in glioblastoma using contrast-enhanced imaging [16]. Due to this disruption, the GBM microenvironment is replete with both resident and infiltrating innate and adaptive immune cells. These lesions are mainly filled with suppressive macrophages and T regulatory lymphocytes (Tregs) rather than effector lymphocytes [17], giving them a “cold tumor” phenotype. As such, immunotherapy is a prime and promising means for modulating this tumor immune landscape to mount a more efficient anti-tumoral response. Its potential is further reflected in the growing number of active clinical trials aimed at using immunotherapies in GBM management.

In this review, we will give an overview of the dynamics of CNS immunity and discuss the seminal and contemporary studies that reveal the mechanisms by which these various immunotherapies can affect glioblastoma tumor immunity and progression. These strategies include chimeric antigen receptor T cells (CAR-T), bispecific T-cell engagers (BiTEs), dendritic cell (DC) vaccination, oncolytic virotherapy, checkpoint blockade, monoclonal antibodies, immune modulators, and lymphatic modulation.

## 2. Immune Activation and CNS Privilege

After taking up self or foreign (including tumor) antigen, DCs—the most effective professional antigen-presenting cells (APCs)—migrate from the tissue site to the accompanying draining lymph node (LN) via afferent lymphatic vessels (LVs). Once in the LN, these antigen-bearing DCs will present to naïve T lymphocytes (T cells) through major histocompatibility class type I (MHC I) or MHC type II (MHC II) complexes. Based on the type of antigen, the levels of co-stimulatory molecules expressed on the APC, and the milieu of the cytokine signals present at the time of priming, either a tolerogenic or an immunogenic T cell response will be initiated. Once activated, these T cells will expand and then recirculate back to the tissue site to reencounter their antigen, where they will then execute their effector functions (i.e., clearance of pathogen or tumor cell via the release of cytokines and cytolytic degranulation). In addition to DCs, other professional APCs at the same tissue site, namely resident macrophages, also house the machinery to efficiently phagocytize and present antigen to T cells, thereby acting as essential shapers of immunity in situ.

Unfortunately, tumors have evolved to evade this system of immune surveillance and activation, using several mechanisms that prevent robust and sustained T cell responses [18]. Furthermore, gliomas have a particularly well-suited environment for growth, given the brain has been long considered a tolerogenic and immuno-privileged organ. This notion stems in part from the following: formative studies showing the lack of rejection of sarcoma tumor and skin transplants placed into the brain parenchyma [19,20], but clearance of the graft if first placed and primed in the periphery [19]; the presence of the tightly controlled BBB; and the lack of DCs or LVs within the CNS parenchyma, unlike in peripheral tissues. However, over the years, we have expanded our understanding of the unique organization of CNS immunity and its surveillance, primarily from studies of models of CNS autoimmunity. Through this work, it is now well recognized that this central and peripheral estrangement is not really the case.

## 3. CNS Lymphatics

Thoughts on how immune surveillance and antigen presentation can occur in such a relatively immuno-privileged organ extended from studies examining fluid and solute homeostasis in the CNS. The brain is one of the most metabolically active organs, constitutively secreting cerebrospinal fluid (CSF) from the choroid plexus, and neurotransmitters, hormones, and trophic factors from neurons and glial cells. Furthermore, interstitial fluid (ISF) is continuously filtered through CNS capillaries, circulating throughout and washing over the brain parenchyma. As such, efficient methods are needed to transport and remove these fluids and metabolic waste from the CNS, in order to maintain a homeostatic balance.

In the periphery, this trafficking of fluids, solutes, waste, as well as antigens and surveilling immune cells is achieved through the lymphatic system [21], which as mentioned above, is not found within the CNS parenchyma. Despite this, through the use of dye and solute tracing studies, various mechanisms for non-LV mediated clearance routes have been described. These mechanisms include CSF drainage into dural sinuses through arachnoid granulations, drainage of CSF along olfactory nerves to the nasal mucosa [22], and the flow of CSF-ISF contents through the brain along Virchow-Robin (i.e., paravascular) spaces via astrocytic aquaporin 4 channels—now designated as the “glymphatic system” [23,24]. Notably, all of these pathways are capable to some degree of draining fluids and soluble CNS antigens into the extracranial superficial and deep cervical LNs [23,24,25], demonstrating a direct connection between the brain and peripheral immune system (Figure 1). The brain then, while lacking parenchymal LVs, developed a pseudo-lymphatic network, with CSF and ISF acting as draining lymph.

While postulated and preliminarily demonstrated in the past [26], how we thought about CNS fluid balance and CNS immune surveillance and response was re-examined following the publication of two studies definitively mapping functional lymphatic vessels in the mouse brain. These LVs are located next to the highly vascular meninges, situated along the dorsal dural sinuses. They contain both MHC II expressing APCs and T cells [27] and were also shown to drain into the deep cervical LNs [27,28]. A later report, using lymphatic reporter mice, MRI, and fluorescent imaging, traced these same meningeal LVs to the base of the skull, where they were demonstrated to have a higher capacity for CNS drainage than the dorsal LVs [29].

In summary, while relatively inert in the steady-state, it is clear that the brain is still capable of initiating immune activation when faced with an insult, as is seen following parenchymal damage, microbial infection, or autoimmune disease flares. This response is mediated by a direct connection between the CNS and the peripheral lymphoid organs, as the initiation of effector responses is lost if cervical LNs are ligated [30,31,32]. Moreover, there are levels of variation depending on the location of the inciting event, as seen with the rejection of tumors and tissue grafts placed in or near the ventricles compared to in the parenchyma [33,34,35]. The brain has thus managed to develop a delicate balance between maintaining a protective environment during the steady-state and being ready to trigger a vigorous response when necessary. It is therefore important to recognize that this CNS privilege relates to both peripheral immunity and brain region—with the parenchyma itself having the most and the “border zones” (including the meninges and choroid plexus) with the least [36].

Understanding the functional organization of CNS lymphatics and antigen drainage is one component of understanding how to approach reinvigorating CNS antitumoral immunity. Another element that we will discuss in the next sections is understanding the factors mediating the immunoregulatory nature of the CNS microenvironment.

## 4. Immune and Stromal Landscape in GBM

### 4.1. Blood–Brain Barrier

All of the major CNS cell types contribute to the overall anti-inflammatory microenvironment of the brain—a sensible evolutionary constraint given the high vulnerability of neurons to irreversible injury from uncontrolled inflammation. The first of these players are the cells that make up the BBB, the CNS vasculature composed of non-fenestrated endothelial cells (ECs) connected by tight junctions (Figure 1). Pericytes and astrocytic end-foot processes surround these ECs and act to give support and integrity to the BBB [37]. The BBB functions as a physical and chemical barrier, actively regulating the trafficking of blood-borne molecules, pathogens, and immune cells into the brain parenchyma [38,39]. There is very limited para- and transcellular flow, and this gatekeeping of what enters the CNS contributes to the difficulty of drug penetrance and the reduced efficacy of many therapies.

Compared to peripheral ECs, the BBB’s ECs have lower expression of adhesion molecules [40], preventing excessive immune extravasation into the CNS. Moreover, these ECs can physically interact with circulating T cells and can modify their activation state depending on the circumstances. Binding of cluster of differentiation (CD) 80 (CD80) (also known as B7-1) and CD86 (also known as B7-2) to CD28 is necessary for the initial activation and expansion of naïve T cells. However, while some have reported that ECs express these molecules in vitro [41,42], recent sequencing of human and mouse brain ECs show little to no basal expression [43,44]. Despite this, ECs do express other co-stimulatory molecules (like CD58, CD40) and have the machinery for processing and presenting self-antigen to T cells on MHC I and MHC II [45]. While the insufficient co-stimulation and cytokine signaling of steady-state ECs would most likely act to tolerize any interacting naïve T cell, they can still modulate an already primed and activated T cell [46,47]. This is in line with evidence that activated, but not naïve, T cells can extravasate and surveil the brain, even in the steady-state [48,49]. This extravasation, although incompletely understood, most likely occurs through transcellular diapedesis, circumventing the need to breach the tight junctions of the BBB [50].

The BBB is often compromised to some degree at the lesion site, contributing to the increased blood-derived immune cells into the tumor [51,52]. Additionally, a compromised and activated BBB is capable of highly upregulating the co-inhibitory molecules programmed cell death ligands 1/2 (PD-L1/PD-L2), which are potent inhibitors of activated T cells. This upregulation occurs in the presence of inflammatory cytokines, particularly interferon-gamma (IFN-γ) and tumor necrosis factor-alpha (TNFα) [53,54]. Importantly, glioma cells can secrete vascular endothelial growth factor (VEGF) A (VEGF-A), a growth factor that acts on ECs to promote proliferation and tumoral neo-angiogenesis [55]. Reciprocally, activated ECs can then secrete additional growth factors that promote glioma stem cell maintenance and tumor development, including transforming growth factor-beta (TGF-β), fibroblast growth factor (FGF), and EGF.

Due to its particularly important role in limiting CNS entry, targeting the BBB would be an essential means of enhancing drug or biologic penetration. The most studied method to enhance BBB penetrance is standard radiation therapy, which not only acts directly on tumor cells but also increases peripheral immune infiltration due to cytokine release from activated endothelial and stromal cells. An additional benefit of radiation therapy is that it can also induce tumor neoantigens, which contributes to the enhanced anti-tumoral responses of radiotherapy when combined with checkpoint blockade [56]. Other methods that can enhance BBB penetrance to therapies include the use of focused ultrasound, convection-enhanced or nanoparticle drug delivery, as well as small molecule inhibitors against EC junctional proteins and efflux pumps—the latter of which mediates the active exclusion of large compounds from entering the brain [51].

### 4.2. Glia and Neurons

If circulating T cells do end up passing the BBB, they will then encounter another challenge. Following extravasation, infiltrating leukocytes would enter into the perivascular space encountering the glia limitans, the final barrier before entering into the parenchyma that is composed of astrocytic end-feet surrounding the BBB ECs. The glia limitans constitutively expresses FasL (also known as CD95 L) [57], a potent inducer of apoptosis once bound to its receptor Fas, which is highly expressed on activated T cells. While a critical component of the BBB, astrocytes are crucial to maintaining the anti-inflammatory milieu of the steady-state CNS, mainly through their secretion of the cytokine TGF-β [58], which is diffusely expressed in the brain and acts to suppress activated T cells—either directly or through its promotion of Treg cell formation. Neurons themselves are also key contributors to this suppressive milieu, as they too secrete TFG-β and other suppressive factors like vasoactive intestinal peptide (VIP) [59]. Furthermore, neurons constitutively express CD200, which directly binds the receptor CD200R to inhibit the activation of various myeloid cells and lymphocytes, including microglia and T cells [60,61]. As such, there is very little presence of T cells directly within the brain parenchyma, with the majority of their surveillance occurring in the perivascular space [58,62].

Receptor activator of nuclear factor kappa B ligand (RANKL) and other such factors secreted from tumor cells help to polarize non-transformed astrocytes, inducing them to produce pro-tumorigenic cytokines (TGF-β, IL-10, IL-6, and insulin-like growth factor 1 (IGF-1)) that enhance tumor growth and infiltration [63,64]. Interestingly, astrocytes are unique in that they can act as critical regulators of tumor progression by forming gap junctions with glioma cells via the molecule connexin 43 [65,66]. In doing so, they can directly provide metabolic and trophic support for tumor growth. Some studies have also demonstrated that activated tumor-associated astrocytes can act like cancer-associated fibroblasts to physically surround a glioma lesion—performing a somewhat aberrant display of normal scar formation and wound healing [67] and also highly expressing the immunomodulatory PD-L1 molecule. As such, this phenomenon of astroglial scarring within the tumor microenvironment (TME) could potentially be contributing to the physical exclusion of activated T cells from these tumors, and thus promoting the “cold tumor” phenotype of malignant gliomas [64,68].

### 4.3. Resident Microglia and Monocyte-Derived Macrophages

Another of the key players of CNS immunity are microglia, the yolk-sac derived resident macrophages of the brain, whose survival is dependent on signaling through colony-stimulating factor 1 (CSF-1) receptor (CSF-1R) either from CSF-1 or IL-34 binding [69,70,71]. Microglia are the only leukocytes within brain parenchyma during the steady-state and act as both immune sentinels and homeostatic regulators—clearing neuronal apoptotic debris and pruning weak synapses throughout brain development [72]. At baseline, microglia have constitutively low expression of MHC II, making them comparatively less efficient at priming and activating T cells than DCs. Even more, microglia have been shown to actively suppress T cell proliferation [73] and inhibit their activation through the secretion of the immunomodulatory cytokines indoleamine pyrrole-2,3-dioxygenase (IDO) and TGF-β [74]. Additionally, neuronal TGF-β has also been shown to suppress microglial activation [75], further contributing to the overall anti-inflammatory milieu of the CNS.

Both animal and clinical studies have shown that GBM lesions are replete with both resident microglia and monocyte-derived macrophages (monoMacs), which also highly express CSF-1R. Together these TAMs have been shown to encompass up to 30% of the cells within the GBM TME. It is also well established that CSF-1R and tumor-associated macrophage (TAM) expression correlate with poorer patient outcomes across various cancers, including malignant gliomas [76]. Macrophages, including microglia, are remarkably plastic cells and adapt quickly to their environment. There is extensive crosstalk between these myeloid cells and the tumor, as the glioma cells release chemokines and cytokines like CSF-1 to draw in and polarize microglia and monoMacs. These polarized TAMs release metalloproteinases, interleukins, and growth factors that further promote glioma growth and infiltration [77].

This suggests that TAMs are a crucial target for GBM therapy and has led to the development and characterization of CSF-1R antibodies or small molecule inhibitors in various preclinical models. Blocking CSF-1R signaling can either deplete or “re-educate” TAMs [78,79]. This “re-education” shifts them toward an anti-tumorigenic phenotype and results in a decrease in the expression of tumor growth factors, a decrease in tumor size, and an increase in overall survival [78,79], suggesting CSF-1R signaling acts in an immunoregulatory capacity. Interestingly, groups have shown that in the brain and peripheral tumors, the survival of macrophages in the presence of CSF-1R inhibition necessitates the presence of factors like granulocyte-macrophage colony-stimulating factor (GM-CSF) [78,80]. These and other studies have prompted investigation into the effects of anti-CSF-1R monotherapy on solid tumor progression, including GBM (NCT01804530) [81]. Unfortunately, results in GBM have shown little improvement compared to control [82]. Work from the Joyce group, however, has provided elucidation of some of the microenvironmental factors potentially mediating this lack of response – particularly TAM derived IGF-1 [67].

It is now also becoming increasingly clear that macrophage ontogeny (i.e., resident embryo derived versus bone marrow derived) might be an essential factor in their functional roles within tissues. This concept was nicely demonstrated in Bowman et al., one of the first to fate map macrophages in a glioma tumor model [83]. This critical concept has since been replicated in other tumor models, including pancreatic and lung adenocarcinoma [84,85]. Moreover, recent studies have shown a select spatial distribution of the resident and blood-derived macrophages within tumor lesions, which has been observed in both human and mouse GBM [86,87,88]. Specifically, these studies show that resident microglia tend to be located in the tumor periphery, while monocytes and monoMacs are more likely to reside within the tumor core. Furthermore, patients with malignant gliomas enriched with a monoMac gene signature have a significant reduction in survival compared to those enriched with a resident microglial signature [86]. Finally, gene profiling of these two distinct macrophages, suggest that it is actually the monoMac compartment that is crucial for driving tumor growth and survival in more aggressive disease, whereas the microglia provided more of a means for tissue infiltration – highlighting the essential functional distinction between these two TAMs.

Current and future work should be aimed at targeting the recruitment of monoMacs via C-C motif (CC) receptor 2 (CCR2) and CC ligand 2 (CCL2) inhibitors, key regulators of monoMac trafficking [89,90]. Furthermore, research should also be focused on the direct inhibition of this critical inflammatory monoMac population via antibodies or small molecules that target proteins that are unique and/or highly expressed on these cells. Some of these now well-established TAM specific proteins include macrophage receptor with collagenous structure (MARCO), osteopontin (SPP1), glycoprotein nonmetastatic melanoma protein B (GPNMB), CD169, apolipoprotein E (APOE), mannose receptor C type 1 (MRC1), and CD73 [77,83,91]. There is potential for additive or synergistic effects when combining the above macrophage targets with checkpoint inhibition, as the former would remove part of the suppressive niche and provide an optimal environment for the T cells (re-activated by checkpoint blockade) to infiltrate and act.

## 5. Preclinical Models of GBM

There are several models used to try to characterize the relevant microenvironmental factors mediating or hampering effective GBM tumor immunity. Some have argued that, in terms of tumor biology, the use of either human (U87MG, U251) or syngeneic mouse (GL261) cell lines is not an accurate representation. This is due in part to their propagation and differentiation in cell cultures. Furthermore, these lines tend to harbor mutations that are not often associated with patient GBM lesions [92]. Instead, many now advocate for moving away from cultured cell lines towards patient-derived xenografts and organoids, as both retain mutations unique to the patient and also retain a sizable glioma stem cell pool. The latter is particularly important for studying tumor initiation and relapse [92,93].

Organoids are derived from the growth and aggregation of patient tumor-initiating cells. They allow for the 3D analysis of the TME and effectively recapitulate the hypoxic and necrotic variation found in human GBM [94]. In addition, they facilitate high-throughput screening of drug therapies. Orthotopic xenografts models provide another type of advantage to 3D organoids, as they are grown within their appropriate tissue environment. These models allow for the necessary crosstalk amongst tumor and non-tumor cells that occur in the natural history of GBM growth. However, the major disadvantage to using patient-derived xenografts is their requirement of an immunocompromised host for their growth, removing the presence of functioning immune cells—a crucial component of the GBM microenvironment.

While it is true that implantation of cell lines does not have exact fidelity to primary tumors, some are readily used pre-clinically, as they are still helpful in studying the importance of microenvironmental influence on GBM progression. This is particularly true for the mouse line GL261, due to its ease of scalability in experiments, presence of key histopathological characteristics of producing pseudo-palisading necrosis and microvascular proliferation. This cell line presents key aberrant signaling pathways (p53, RAS), presence of cells that exhibit stem qualities (CD44+, CD133+), its “partial immunogenicity” that mimics well the “cold tumor” phenotype of human GBM, and most critically, its ability to be studied in immunocompetent in vivo systems [95,96,97,98]. Furthermore, GL261 seems to be an excellent model to study the formerly mentioned mesenchymal subtype of GBM [97], as it is filled with TAMs that highly express *Gpnmb* and *Spp1,* which are genes highly associated with human mesenchymal GBM tumors compared to other subtypes [99].

Along these lines, another widely used preclinical system is the genetically engineered mouse model (GEMM). GEMMs use specific glioma-associated germ-line mutations in conjunction with breeding strategies to generate tumor-inducing events under tight spatial and temporal control [100]. These systems include CRE inducible systems [101] as well as replication-competent avian sarcoma leukosis virus long terminal repeat with a splice acceptor-tumor virus A (RCAS-TVA) systems [102], whereby retroviral vectors are used to introduce defined mutations into specific cells expressing the virus’s cognate receptor [103]. The advantage of these genetic models is that they permit the study of discrete GBM subtype pathophysiology and therapy response—for example, RCAS-TVA-PDGFRa (proneural subtype) [78,104] and Cre-Lox expression of EGFRvIII or RCAS-TVA-EGFR (classical subtypes) [102,105].

Even more complex models expressing key combinations of driver mutations that form these various GBM subtypes may soon emerge, as it is now clear that the expression of these mutations is formative in shaping the tumor immune landscape [106,107,108]. Even more importantly, the true significance of these gene mutation-immune signatures may lay in the patient treatment response, following stratification.

## 6. Enhancing the Adaptive Immune Response in GBM

We have already established some of the main innate components contributing to the immunosuppressive nature of the GBM TME that need to be addressed to promote ideal anti-tumoral effector responses. Here, we now discuss some of the most promising methods for supporting the adaptive arm of the tumor immune response.

It is well-known that there is a marked depletion (and aberrant function) of effector T cells in the blood and at the tumor site of patients with GBM, contributing to the overall immune suppression and evasion often seen in many cancers [109]. Interestingly, a mechanism for this GBM induced T cell reduction was not established until recently [110]. Chongsathidkiet et al. demonstrated that, in both mice and patients with GBM, this T cell-specific lymphopenia was, in fact, due to sequestration of naïve T cells within the bone marrow. This process depends on the internalization of the G-protein coupled receptor sphingosine-1-phosphate receptor 1 (S1P1) on the T cell surface, signaling that it is critical for proper T cell trafficking to tissues and lymphoid organs. As such, many of the following methods will address this particular suppressive phenomenon, i.e., increasing T cell infiltration and/or enhancing the quality of the T effector cell response in tumors.

### 6.1. Oncolytic Viral Therapy

Observations of tumor regression in patients with concurrent viral infections have occurred for over a century [111,112]. Subsequently, there have been many preclinical and clinical studies trying to exploit oncolytic viruses as a therapeutic strategy for various cancers. Initially, many thought the main anti-tumoral effect of these viruses was solely via virus-induced tumor cell death. However, the major power of oncolytic viruses lies in their ability to robustly activate the immune system via the release of tumor-specific antigens and damage associated molecules [113,114,115]. This ability to change the TME makes them potentially powerful agents when concurrently used with other forms of immunotherapy, like checkpoint blockade [116]. Various types of oncolytic viruses are currently being studied, including herpes, poliovirus, adenovirus, cytomegalovirus (CMV), measles, coxsackie, parvovirus, and reovirus [113,117,118,119]. Historically, much work was done using herpes viruses as oncolytic therapy for malignant glioma [120]. However, recent promising results have emerged from phase I and II clinical trials using poliovirus and CMV [121,122,123].

### 6.2. DC Vaccination

Another well-established method to enhance the quality of T cell priming and effector response is DC vaccination. As previously discussed, DCs are the most effective activators of T cell proliferation and effector function. Their presence is a requirement for anti-tumoral immunity, either with or without checkpoint blockade [124,125]. Many groups have taken advantage of this by sorting DCs directly from the blood or differentiating them from blood monocytes or bone marrow progenitor cells using GM-CSF and IL-4, before pulsing them with either whole tumor lysate or isolated tumor antigen (single or multiple peptides, tumor DNA or RNA). These tumor antigen-loaded DCs are then administered back into the host, either systemically or in situ. They then migrate to meet and robustly activate T cells in the draining LNs or locally at the tumor site. Several pre-clinical studies have shown this to be an effective means of increasing effector CD8 T cell infiltration into glioma, and of increasing survival outcomes [126,127,128].

However, while promising in animal models, clinical trials of DC vaccines have yet to show a significant increase in overall survival [129]. Despite this, the results of these trials have established that DC vaccines have an excellent safety profile and provide some evidence that there was an initiation to some extent of cellular and humoral response following administration [129]. Given this, many have tried to boost DC stimulation in various ways to improve the immune response and clinical outcome. For example, one group reported pre-conditioning the vaccination site with a potent antigen to induce local inflammation before vaccination administration [130]. Doing this increased antigen-loaded DC migration to the LNs, and improved tumor burden and survival [130]. Meanwhile, other groups have tried to improve DC efficacy with the use of local induction of FMS-related tyrosine kinase 3 (FLT3), whose signaling is key to the recruitment and expansion of DCs at the tumor site [131]. Adjuvants, such as chemotherapy or toll-like receptor (TLR) agonists, are another means for boosting DC maturation, as they are potent DC activators and facilitators of anti-tumoral response, particularly in the context of checkpoint blockade [125,132,133].

### 6.3. Checkpoint Inhibition

Checkpoint inhibition has changed the course of cancer therapeutics, particularly advanced metastatic melanoma [134]. Although now trialed across a wide variety of solid tumors, only about 20 to 30% of patients have an enduring response with checkpoint blockade [135]. It is urgent to address the suppressive mechanisms both within and outside of the TME responsible for the low rates of responses. Checkpoint molecules, including cytotoxic T lymphocyte-associated protein 4 (CTLA-4) and PD-1, are upregulated upon T cell activation and normally act to suppress the immune response and limit prolonged inflammation and damage at the tissue site following resolution of an insult. The cognate ligands to these molecules are highly expressed on other T cells (with CTLA-4 competing with B7 co-stimulation for CD28 binding) on APCs (with PD-L1 on DCs and macrophages), or on the tumor cells themselves (primarily PD-L1). When these molecules are inhibited, this “immune break” is lifted, and allows for continued unencumbered T cell receptor co-stimulation and effector response. As such, high histological expression of PD-L1 on tumor lesions is now being applied as an indicator for checkpoint blockade use for many cancers [135]. In GBM specifically, there have been many preclinical demonstrations of the effectiveness of checkpoint inhibition [136,137], showing reduced tumor burden, increased CD8 effector cell infiltration, and/or reduced Treg cell infiltration, depending on the agent administered. Despite this preclinical data, little improvement has been seen clinically [129,138]. However, combinatorial strategies of chemotherapy with checkpoint blockade have shown particularly promising means of circumventing this lack of clinical response in other tumors [139].

### 6.4. CAR-T Cell Therapy

Another route to activating a robust and antigen-specific adaptive response is to genetically engineer T cells whose sole targets are tumor antigens—as seen in CAR-T cell therapy. In GBM, this method has been used in a subset of patients with IL-13Rα and human epidermal growth factor receptor 2 (HER2) mutations; however it has been most extensively studied in the subpopulation of patients with EGFRvIII mutations [140,141,142,143,144]. While preclinical and clinical data does show that GBM CAR-T cells can induce tumor infiltration, the survival benefits have been limited. This method tends to induce heavy selective pressure onto tumor cells that do not express the targeted antigen – often leading to immune escape, tumor relapse, and disease progression. This particular phenomenon is less of an issue with clonal hematologic malignancies, like lymphoma, where CAR-T cells have outstanding efficacy [145]. Combinatorial CAR-T cell strategies, using several antigen targets, will need to be developed in the future in order to circumvent this selective tumor antigen escape phenomenon.

### 6.5. BiTEs

Another form of T cell “re-education” and “re-direction” is through the use of BiTEs, which are two linked recombinant monoclonal antibodies that target both a T cell surface molecule (most often CD3) and a tumor-specific surface antigen [146,147] – linking the T cell to the tumor cell. In this way, T cells can bind and engage with tumor cells outside of the classical MHC:T cell receptor synapse. While BiTEs have most traditionally been studied in the context of leukemia and other hematologic malignancies [148], one of the more highly studied tumor antigens targeted by BiTEs in malignant glioma is EGFRvIII [148,149,150]. Unlike CAR-T cells, BiTEs allow for a polyclonal T cell response, which can be advantageous in helping to reduce the likelihood of tumor antigen escape. However, one key disadvantage of these T cell engagers is that continued administration of the recombinant antibodies is necessary to maintain an immune response and to prevent tumor recurrence [146,147]. As such, while very promising, BiTEs require more work to improve their current level of efficacy and to maintain a more sustained anti-tumoral response.

### 6.6. CNS Lymphatic System Modulation

Lastly, another interesting method to boost T cell infiltration and enhance checkpoint response is the modulation of the CNS lymphatic vessels. As discussed previously, LVs are essential for fluid circulation and immune surveillance in any tissue. The recent discovery of functional LVs that drain into the deep and superficial cervical LNs within the CNS has significant implications for antigen flow and tumor response [24,28]. This is particularly intriguing given that some have shown there is a decrease in CSF and CNS antigen flow to the draining LNs [151], as well as defects in the transcriptional signature, morphology, and function of the meningeal lymphatic vessels in pre-clinical models of glioma [152,153]. These deficiencies subsequently result in a reduction of CNS DCs migrating into the tumor LNs from the tumor site. These groups demonstrated that VEGF-C administration, a critical lymphatic growth factor can overcome these suppressive features [152,153]. Specifically, overexpressing VEGF-C expanded CNS LVs, mostly in the dorsal aspect of the brain. The anti-tumoral response was higher than the immune checkpoint inhibitor alone when VEGF-C was added to the immune checkpoint blockade. As proof of concept, this anti-tumoral response was abrogated when these dorsal LVs were ligated [153]. Additionally, virally overexpressed VEGF-C resulted in the rejection of tumor growth, in both primary and metastatic glioma models, an effect mediated by increased tumor-specific CD8+ T cells. This positive correlation between T cell infiltration and VEGF-C expression was also found in patients treated with checkpoint inhibition. Given these results, modulating the lymphatics before the immune checkpoint blockade would be an exciting and promising strategy in future clinical trials.

## 7. Conclusions and Perspectives

Despite the wealth of knowledge gained regarding its molecular and cellular origins and its pathophysiology, effective GBM therapy remains elusive. However, recent advances made in the deep characterization of the TME have cleared new ground for assessing novel and effective treatments for this fatal disease. It has become more than apparent that the suppressive TME plays an enormous part in mitigating anti-tumoral immunity. Within the TME, the myeloid compartment is a major area of study, particularly given the enormous presence of immunosuppressive TAMs. This suppressive TME is further compounded by active sequestration of T cells from the blood and tissue, with the few effectors displaying an exhausted phenotype. Some of the more extensively studied exhaustion markers include inducible T cell co-stimulator (ICOS), lymphocyte activating 3 (LAG3), and T cell immunoglobulin and mucin domain-containing protein 3 (TIM3) [109]. Exhausted T effector cells are found within the tumor, both with and without the use of checkpoint inhibition. While potentially reversible, T cell exhaustion is a major problem that needs to be overcome for sustained anti-tumoral responses. Thankfully, there is already work suggesting that combinatorial inhibition of these select molecules will be effective in the reversal of T cell exhaustion [154].

The ideal treatment strategy to generate robust anti-tumoral responses will likely come from the combination of a standard of care therapy along with a multi-pronged immunological approach: depleting/re-educating TAMs, boosting neoantigen expression with the use of radio- and chemotherapy, enhancing DC migration and activation in an antigen-specific manner, and finally enhancing T cell effector function—either by taking off their breaks with a checkpoint, engineering them with a specific antigenic target, or aiding their recognition and binding of tumor cells.

As implied throughout this work, GBM therapy should quickly shift towards a “personalized” route, as we continue to stratify patients on both a molecular and microenvironmental level. Even more, continued immune and stromal cell monitoring of tumors—specifically in patients who do not respond to immunotherapy—will be crucial in revealing the high yield targets that need to be hit in order to overcome therapy resistance.

## Figures and Tables

**Figure 1 ijms-21-07358-f001:**
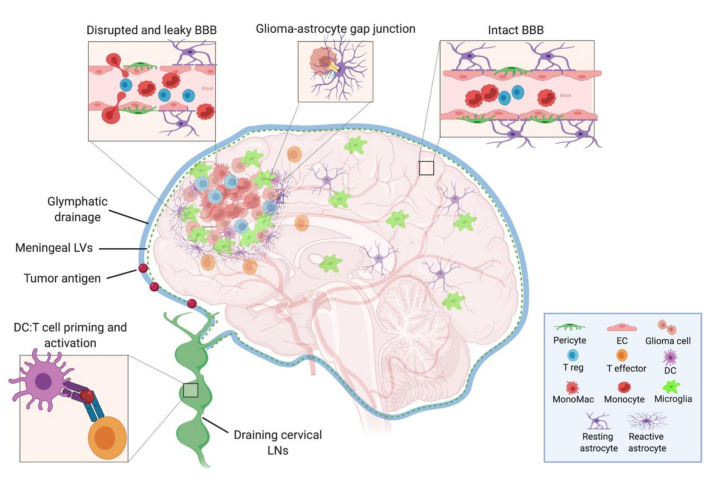
Model depicting the cells of the intact and leaky BBB, the cellular components of the neuroimmune system in the brain, and the tumor microenvironment. The CSF, meningeal lymphatic, and glymphatic fluids drain to the deep cervical LNs, where DCs that have captured either CNS or tumor antigens can present to and activate naïve T cells. Within the tumor, there are suppressive Tregs, monocytes, monocyte-derived macrophages, microglia, and reactive astrocytes—the latter of which may act in an aberrant wound healing manner to wall off the lesion and inadvertently exclude effector T cell from entering. BBB: blood–brain barrier, CSF: cerebrospinal fluid, LN: lymph node, DC: dendritic cell, EC: endothelial cell, LV: lymphatic vessel. (Created with BioRender.com).

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
