# Peer review of "The CNS and the Brain Tumor Microenvironment: Implications for Glioblastoma Immunotherapy"

_ijms, 2020, doi:10.3390/ijms21197358_

Round 1
Reviewer 1 Report
The review is useful and gives important and concise information, useful vor oncologists, GBM researchers, students and general readers of cancer biology and therapy. However, small changes need to be done.
Abstract: The abstract is too general and does not give any specific information. Could be used to any cancer. Thus, should be re-written in order to give a real resume of the contents of the article.
Review: As written, the review is only useful for people that already knows the information. Most acronyms are not defined neither their importance in the Physiopathology of GBM, as, for instance, line 40, IDH1/2 and many others before and after.
In this way, from lines 191 to 199, regarding BBB penetrance modified by radiation therapy or some “molecule inhibitors to EC” should be also clarified. Also Ab´s or small molecules… in lines 288 and 289. 386 – induction of Flt3… (459) ICOS, LAG3, and TIM·
Author Response
- The review is useful and gives important and concise information, useful for oncologists, GBM researchers, students and general readers of cancer biology and therapy. However, small changes need to be done.
We thank the reviewer for the criticism of our manuscript and incorporated the reviewer’s suggestions.
- Abstract: The abstract is too general and does not give any specific information. Could be used to any cancer. Thus, should be re-written in order to give a real resume of the contents of the article.
As per the reviewer's suggestion, we modify the abstract to reflect the manuscript's content and be more disease-specific.
- Review: As written, the review is only useful for people that already know the information. Most acronyms are not defined neither their importance in the Physiopathology of GBM, as, for instance, line 40, IDH1/2 and many others before and after. In this way, from lines 191 to 199, regarding BBB penetrance modified by radiation therapy or some “molecule inhibitors to EC” should be also clarified. Also Ab´s or small molecules… in lines 288 and 289. 386 – induction of Flt3… (459) ICOS, LAG3, and TIM
Throughout the manuscript, we defined the acronyms and described the physiopathology of important alterations such as MGMT or IDH. We also define Flt3 and the various exhaustion markers ICOS, LAG3, and TIM3.
Reviewer 2 Report
The review is well structured, well illustrated, comprehensive and clearly written. The authors have made a great effort in summarizing all the literature data available in the field (which is not only broad but also rapidly developing). The topic chosen is of major importance, as glioblastoma unfortunately remains and unresolved clinical issue and continues posing significant therapeutic challenges. This manuscript thus holds the promise of making a nice contribution to the field of cancer immunotherapy. The overall value of the manuscript could be increased by introducing the following minor revisions:
- The abbreviation list should be re-checked. Abbreviations, that some readers, who are not experts in the GBM field in particular may not be familiar with, should be ideally introduced at first mentioning (e.g. MGMT on line 26, PDGFR on line 38, IDH on line 40, BBB on line 88, paragraph lines 180-188, and so on …)
- The authors may consider adding a paragraph on another strategy currently applied in order to break immune tolerance and induce antitumoral immune responses in GBM, namely the oncolytic virotherapy. This approach may be mentioned on lines 63-64, and briefly described in a separate paragraph under the “Enhancing the adaptive immune response in GBM” chapter title. Oncolytic virotherapy is based on immunogenic tumor cell death, TME warming up and antitumor immune response induction. Therefore, this original type of cancer immunotherapy – viroimmunotherapy – deserves at least a mentioning
- Another strategy that is missing a short presentation/mentioning in the paper is the development of bispecific T cell engagers (BiTEs) for glioma treatment. Considering the profile of the International Journal of Molecular Sciences readership, this approach may trigger interest and increase the impact of the review
- Some sentences should be restructured for the sake of clarity (e.g. “Additionally …” on lines 181-184) or revised for spelling mistakes (e.g. lines 193, 195-196, 298-299)
- If the authors agree, they may further subdivide the “Enhancing the adaptive immune response in GBM” chapter into subparagraphs with corresponding subtitles, as in the “Immune and stromal …” chapter: i.e. DC vaccination, Checkpont inhibition, CAR T cell therapy, CNS lymphatic vessel modulation
Author Response
- The review is well structured, well illustrated, comprehensive and clearly written. The authors have made a great effort in summarizing all the literature data available in the field (which is not only broad but also rapidly developing). The topic chosen is of major importance, as glioblastoma unfortunately remains and unresolved clinical issue and continues posing significant therapeutic challenges. This manuscript thus holds the promise of making a nice contribution to the field of cancer immunotherapy. The overall value of the manuscript could be increased by introducing the following minor revisions:
We appreciated the reviewer’s comments and revised according to the reviewer’s suggestions.
- The abbreviation list should be re-checked. Abbreviations, that some readers, who are not experts in the GBM field in particular may not be familiar with, should be ideally introduced at first mentioning (e.g. MGMT on line 26, PDGFR on line 38, IDH on line 40, BBB on line 88, paragraph lines 180-188, and so on …)
We defined the abbreviations throughout the manuscript, and we believe it is much clearer.
- The authors may consider adding a paragraph on another strategy currently applied in order to break immune tolerance and induce antitumoral immune responses in GBM, namely the oncolytic virotherapy. This approach may be mentioned on lines 63-64, and briefly described in a separate paragraph under the “Enhancing the adaptive immune response in GBM” chapter title. Oncolytic virotherapy is based on immunogenic tumor cell death, TME warming up and antitumor immune response induction. Therefore, this original type of cancer immunotherapy – viroimmunotherapy – deserves at least a mentioning.
We agree with the reviewer, and we included a paragraph on oncolytic viral therapy.
- Another strategy that is missing a short presentation/mentioning in the paper is the development of bispecific T cell engagers (BiTEs) for glioma treatment. Considering the profile of the International Journal of Molecular Sciences readership, this approach may trigger interest and increase the impact of the review
We appreciate the reviewer's suggestion and included a paragraph on BiTEs for GBM treatment.
- Some sentences should be restructured for the sake of clarity (e.g. “Additionally …” on lines 181-184) or revised for spelling mistakes (e.g. lines 193, 195-196, 298-299)
We appreciate the reviewer finding text flow issues and have re-worded the referenced lines for more clarity. Additionally, while we appreciate the reviewer pointing out spelling mistakes, we could not find them on the mentioned lines – we would be happy to revise if the specific spelling errors are highlighted in those references' lines.
- If the authors agree, they may further subdivide the “Enhancing the adaptive immune response in GBM” chapter into subparagraphs with corresponding subtitles, as in the “Immune and stromal …” chapter: i.e. DC vaccination, Checkpont inhibition, CAR T cell therapy, CNS lymphatic vessel modulation
We thank the reviewer for the suggestion and included the subparagraphs with corresponding subtitles.
Reviewer 3 Report
Dear authors,
in the manuscript by Fiona Desland et al., you reviewed the different implications for glioblastoma immunotherapy.
I consider this work quite interesting, and I think it deserves another effort to get better. My comments are as follows:
- In the Abstract, you should make explicit the abbreviations (for example GBM).
- In the introduction, you should better explain the aim of the review. In particular, what does this review add to the state of the art?
- In the Introduction, you report “work in recent years has incorporated tumor gene expression, gene mutation, and epigenetic signatures into a patient’s tumor classification, prognosis, and treatment response” (lines 33-35). You should add RECENT references to support your statement.
- Throughout manuscript, the authors should report the correct and, where possible, even more recent references (For example, line 242, line 87 etc.). Many references reported date back more than ten years.
- The only figure is not clear and the dimension is too big and not fit on the page. Moreover, the authors should add more figures to explain what is reported in the text.
- The conclusions are unclear. You should rewrite them focusing attention on the aim of this review.
Author Response
- in the manuscript by Fiona Desland et al., you reviewed the different implications for glioblastoma immunotherapy.
I consider this work quite interesting, and I think it deserves another effort to get better. My comments are as follows:
We thank the reviewers for the comments and revised accordingly.
- In the Abstract, you should make explicit the abbreviations (for example GBM).
We defined the abbreviations in the abstract.
- In the introduction, you should better explain the aim of the review. In particular, what does this review add to the state of the art?
We took into account the suggestion, and we hope to have detailed the review's aim better.
- In the Introduction, you report “work in recent years has incorporated tumor gene expression, gene mutation, and epigenetic signatures into a patient’s tumor classification, prognosis, and treatment response” (lines 33-35). You should add RECENT references to support your statement.
We appreciated that the reviewer pointed out the missed references. The references are now included (#7-12).
- Throughout manuscript, the authors should report the correct and, where possible, even more recent references (For example, line 242, line 87 etc.). Many references reported date back more than ten years.
We took into account the suggestion and included more recent references (#1, 51, 52, 70-76, 90-93, 109, 112-124, 134, 146-151).
- The only figure is not clear and the dimension is too big and not fit on the page. Moreover, the authors should add more figures to explain what is reported in the text.
Although the figure fits our display on the page, we reduced its size, and we hope that helped make it clear.
- The conclusions are unclear. You should rewrite them focusing attention on the aim of this review.
We re-wrote the conclusions, and we hope that it better highlights the aim of the review.
Round 2
Reviewer 1 Report
The authors made the changes suggested, thus, from my poit of viw, the manuscript should be published.
Reviewer 3 Report
Dear authors,
You responded to each point of my comments and modified the manuscript according to my suggestions where possible. In this new version, I consider this work more interesting and for this reason, I think that this manuscript should be accepted to be published in this journal.